# Signaling Mechanisms and Pharmacological Modulators Governing Diverse Aquaporin Functions in Human Health and Disease

**DOI:** 10.3390/ijms23031388

**Published:** 2022-01-26

**Authors:** Kim Wagner, Lucas Unger, Mootaz M. Salman, Philip Kitchen, Roslyn M. Bill, Andrea J. Yool

**Affiliations:** 1School of Biomedicine, University of Adelaide, Adelaide, SA 5005, Australia; kim_wagner@gmx.de; 2College of Health and Life Sciences, Aston University, Birmingham B4 7ET, UK; l.unger@aston.ac.uk (L.U.); p.kitchen1@aston.ac.uk (P.K.); 3Department of Physiology Anatomy and Genetics, University of Oxford, Oxford OX1 3QX, UK; mootaz.salman@dpag.ox.ac.uk; 4Oxford Parkinson’s Disease Centre, University of Oxford, South Parks Road, Oxford OX1 3QX, UK

**Keywords:** aquaporin (AQP), membranes, water, fluid, secretion, osmosis, facilitated diffusion, signaling

## Abstract

The aquaporins (AQPs) are a family of small integral membrane proteins that facilitate the bidirectional transport of water across biological membranes in response to osmotic pressure gradients as well as enable the transmembrane diffusion of small neutral solutes (such as urea, glycerol, and hydrogen peroxide) and ions. AQPs are expressed throughout the human body. Here, we review their key roles in fluid homeostasis, glandular secretions, signal transduction and sensation, barrier function, immunity and inflammation, cell migration, and angiogenesis. Evidence from a wide variety of studies now supports a view of the functions of AQPs being much more complex than simply mediating the passive flow of water across biological membranes. The discovery and development of small-molecule AQP inhibitors for research use and therapeutic development will lead to new insights into the basic biology of and novel treatments for the wide range of AQP-associated disorders.

## 1. Aquaporin Structure, Function, and Localization

Aquaporins (AQPs) have diverse roles in mammals, ranging from fluid homeostasis, glandular secretions, barrier function, immunity and inflammation, cell migration, and angiogenesis to signal transduction and sensation. It is now clear that AQP functions are more complex than simply mediating the passive flow of water across biological membranes [1]. Understanding their underlying regulatory mechanisms along with the discovery and development of small-molecule AQP inhibitors for use in research and therapeutic development is expected to lead to new insights into the basic biology of and novel treatments for the wide range of AQP-associated disorders. Here, we review the signaling mechanisms and pharmacological modulators governing diverse aquaporin functions in human health and disease. Abbreviations are listed in Abbreviations.

### 1.1. Distribution and Classification of AQPs in the Human Body

The essential role of membrane intrinsic protein channels in the regulation of water transport and homeostasis was discovered in 1986 [2,3]. The molecular characterization of the archetypal water channel protein, aquaporin-1 (AQP1), launched the research field in the early 1990s with the cloning and characterization of CHIP28 (later renamed AQP1 [4,5]). Recognizing earlier work, the lens major intrinsic protein (MIP), cloned in 1984 [6], was renamed AQP0 on the basis that it formed ion channels in lipid bilayers [7] and transported water, albeit at a considerably slower rate than AQP1 [8]. Additional members of the mammalian AQP family were cloned and characterized over the next decade; 13 paralogs have currently been identified in higher mammals, showing differential patterns of tissue and cell expression throughout the body (Figure 1). Their diverse functions include the transport of water, glycerol, gases, hydrogen peroxide (H_2_O_2_), ammonia, and ions [9,10,11,12].

The traditional categorization of aquaporins (AQPs) into orthodox (water-selective) and glycerol-permeable subtypes, with a third poorly understood ‘super’ or ‘subcellular’ group, no longer encompasses the expanding appreciation of AQPs as multi-functional channels. A broad repertoire of permeable substrates is evident not only in mammalian classes but in AQP classes across the kingdoms of life [13]. In the traditional scheme, orthodox mammalian AQPs comprise AQP0, 1, 2, 4, 5, and 6. This group includes the classic water channels, but interestingly also includes all the mammalian AQPs known thus far to have dual roles as water and ion channels (which are AQP0, 1, and 6). No ion channel function has yet been detected for AQP4 and AQP5 [14], though it is possible that the key stimuli remain to be identified, a concept to be considered for any AQPs that appear to be nonfunctional in experimental assays. When a multiple-sequence alignment analysis is run for human AQPs combined with known non-mammalian AQP ion channels, AQP8 falls on a distant branch of the orthodox group alongside the soybean AQP channel nodulin 26, which conducts water, glycerol, ammonia, and ions [1,15,16]. Although AQP8 in other alignments has been assigned to the ‘super-aquaporin’ group with AQP11 and AQP12 [17], its capacity for conducting ammonia [18] presents an interesting similarity with nodulin 26.

AQP0 was the first mammalian water channel suggested to mediate an ion conductance, in addition to its function as a water channel [19,20]. AQP1′s function as a gated ion channel was proposed in 1996 [21], and has since been refined to clarify that AQP1 activation occurs via the direct binding of cyclic guanosine monophosphate (cGMP) [22,23,24]. The subsequent identification of other members of the AQP family as dual water and ion channels has added the mammalian water channel AQP6, the insect water channel *Drosophila* big brain (DmBIB), and plant membrane intrinsic proteins such as the *Arabidopsis* plasma membrane intrinsic protein (PIP2;1) to the list [14,25,26].

The aquaglyceroporin group (AQP3, 7, 9, and 10) captures well-characterized glycerol-permeable channels. The most-recently cloned AQPs from higher mammalian orders include AQP11 and 12 [27]. Initial tests found no water permeability for AQP11 expressed in *Xenopus* oocytes [28]; however, reconstituted in membrane vesicles, AQP11 showed a low but detectable water permeability that was sensitive to mercury [29]. At positions which correspond to the conserved arginine aromatic barrier in most AQPs, residues in AQP11 and 12 are uniquely hydrophobic, lacking both the aromatic and basic elements of this functionally important site. Although this should not be compatible with the transport of water, alternative barrier sites have been proposed [30].

### 1.2. Structural Biology of the AQP Family

Owing to the relative ease of crystallizing AQP proteins, the structural biology of the transmembrane domains of the AQP family is well-established. A conserved signature fold consists of six transmembrane helices. Two re-entrant helix-forming loops stack one on top of the other, with the family’s signature asparagine–proline–alanine (NPA) motif present in both helices at their interface (Figure 2A). In contrast to the transmembrane domains, little is known about the structures of the intracellular amino- and carboxy-termini of AQP proteins. These less-ordered regions are usually removed to facilitate crystallization. Complex formation by calmodulin (CaM) binding to the unstructured C-terminus of AQP4 causes it to adopt an α-helical conformation [31]. The short C-terminal domain of AQP2 exemplifies this flexibility; the C-termini in each monomer adopted four different conformations within the tetrameric unit cell in an X-ray crystal structure [32].

Following a pattern observed for cyclic-nucleotide-gated channels and potassium channels [33], AQP tetramers have a central pore at the four-fold axis of symmetry [34,35,36], which remains incompletely characterized for most AQP classes. Pharmacological and functional analyses have shown in AQP1 that ion and water transport occurs through independent parallel pathways [11].

Despite the ubiquity of tetramerization in the AQP family, the water pores reside and function within monomers, as evidenced by AQP structures in which permeable substrates have been co-crystallized in the intrasubunit pore in single-file [34,37,38,39]. AQP1 fusion proteins, linking one functional and one non-water-conducting monomer into dimers, were used to demonstrate that each subunit contains an independent water pore pathway [40]. In human AQP4, mutations in the loop D domain were shown to reduce oligomerization, impairing membrane trafficking and its responsiveness to osmotic stimuli [41].

The molecular mechanisms of ion permeation and gating differ between AQP classes. In AQP1, cGMP-activated cation currents are thought to flow through the central pore [36]. Molecular dynamic simulations and mutagenesis revealed that cGMP interacts with an arginine-rich region in loop D, causing loop displacement and conformational changes, and widening the central pore to enable hydration and then ion permeation [24,42,43]. In the closed central pore, hydrophobic barriers restrict ion permeation but could allow the permeation of gases, such as CO_2_ [44]. The loop D domain of AQP1, modeled as the gate for cGMP-dependent ion-channel-opening, might interact with the C-terminus, which modulates activation rates [22,23,42,43,45].

Amino acid sequence similarities between the AQP1 C-terminus and other known cGMP-interacting proteins fit a proposed modulatory role for the C-terminal domain [45]. The activation of ion conductance is impaired in AQP1 channels with truncated or mutated C-termini [46]. In contrast, in AQP6 the Hg^2+^-inducible permeation pathways for anions appear to reside in the intrasubunit (monomeric) pores, based on the Hg^2+^-sensitive cysteine locations and the lack of allostery in Hg^2+^-induced activation [47]. AQP6 channel properties are affected by the conformational flexibility of transmembrane helices involving conserved glycines at the crossing point of transmembrane domains TM2 and TM5; the mutation of a glutamine uniquely present in rat AQP6 into a glycine typical of most AQPs (N60G) abolished anion conduction while increasing water permeability [48]. The molecular basis of ion transport through AQP0 remains unexplored.

### 1.3. AQP Permeabilities: An Expanding Repertoire

In contrast to initial expectations, AQPs have turned out to be more than simple water channels, and display properties of a diverse multifunctional protein family that is still not fully characterized. The spectrum of roles now recognized for AQPs include water, glycerol, urea, ammonia, nitric oxide, and H_2_O_2_ transport; ion conductance; direct mediation of cell–cell adhesion; and regulation of the plasma membrane abundance of other membrane proteins (Table 1). Emerging permeability properties do not segregate neatly into the traditional orthodox and aquaglyceroporin classification scheme. Classes of AQPs that facilitate H_2_O_2_ transport, described as peroxiporins, include AQP1 [49], AQP3 [50], AQP5 [51], AQP8 [50,52,53], and AQP9 [54], although earlier work on AQP1 did not find peroxiporin activity [55]. AQP8 is expressed in the inner membrane of mitochondria and involved in H_2_O_2_ transport linked with the accumulation of reactive oxygen species (ROS) [52], revealing an unexpected breadth of physiologically important roles for AQPs across phyla [13,56] and highlighting many gaps in our knowledge that are yet to be addressed. Recent perspectives have provided comprehensive synopses of AQP-related diseases beyond the scope of this review [57,58,59,60,61,62,63,64,65,66,67,68].

The diverse array of AQP functions in human physiology and pathophysiology has been divided into four overarching themes for the purposes of this review. These are: fluid homeostasis and secretion; signal transduction and sensory functions; defense and metabolism; and motility and cancer (Figure 3).

## 2. AQPs in Fluid Homeostasis and Secretion

### 2.1. Water Transport in the Kidneys

A striking example of the importance of water transport mediated by orthodox AQPs is found in the kidneys, which are the principal organs for maintaining whole-body homeostasis by filtering blood, recovering selected solutes, controlling water balance, and excreting waste products in concentrated urine. Among the nine AQPs expressed in human kidneys [92] (Figure 1), AQP2 has been the most intensively studied.

AQP2 is regulated by the pituitary hormone arginine vasopressin. Binding to its native receptor, the vasopressin type 2 receptor (V_2_), results in the G-protein-coupled stimulation of adenylyl cyclase, the production of cyclic adenosine monophosphate (cAMP), the activation of protein kinase A (PKA), and the phosphorylation of the AQP2 C-terminus [93,94]. The cAMP/PKA signaling cascade results in AQP2 translocation to the apical membrane, enhancing water permeability at the luminal side [95,96]. AQP2 mutations or dysfunction lead to severe urinary concentrating defects [97]. Abnormally low levels of or mutations in AQP2 or V_2_ receptors can cause polydipsia, polyuria, and associated electrolyte imbalance, resulting in severe dehydration in conditions such as nephrogenic diabetes insipidus (NDI); conversely, excessive AQP2 levels can induce hyponatremia, leading to heart and liver failure [92]. NDI is characterized by the inability of the kidneys to concentrate urine despite increased vasopressin levels [98]. Genetic alterations in the other renal AQPs are uncommon and have not been found to cause severe outcomes in most cases [99,100].

AQP2, 3, and 4 are located in cells lining the collecting duct [101], which are crucial regions that control body water homeostasis and urine concentration [93]. AQP1 is predominant in the apical and basolateral cell membranes of the proximal tubules and thin descending limb of the loop of Henle, accounting for more than 70% of water reabsorption from filtrate [92,102]. AQP1-deficient mice show a urinary concentrating defect due to reduced water permeability and low fluid absorption in the proximal tubule, thin limbs of the loop of Henle, and vasa recta [103,104]. Striking increases in AQP1 mRNA and protein levels have been observed in glomeruli in diverse nephropathy- and glomerulonephritis-related pathologies [105].

### 2.2. CSF Production by the Choroid Plexus

Tight regulation of central nervous system (CNS) water content is fundamental for the proper functioning of the brain, which is exquisitely sensitive to increased intracranial pressure (ICP) [106]. Due to the rigid skull, edema and an associated build-up of pressure in the brain can quickly turn into life-limiting conditions via compression, ischemia, and consequent cell death [107]. AQPs described in the CNS thus far include AQP1, 3, 4, 5, 8, and 9, with AQP1, 4, and 9 dominating the research in the field [107]. AQPs in the CNS have an essential role in the production, circulation, and clearance of the cerebrospinal fluid (CSF) needed for removing waste products, carrying signaling molecules, and maintaining the homeostatic electrolyte balance [108], which is important for proper neuronal excitation [109,110].

In the brain, AQP1 is expressed in the apical choroid plexus (lining the ventricles), where it facilitates high fluid transport rates for CSF secretion [111,112]. CSF production is influenced by the regulation of Na^+^/K^+^ ATPase pump activity and carbonic anhydrase levels [113]. The overproduction of CSF could exacerbate fluid accumulation in the injured brain, based on observations that CSF production was decreased by approximately 25% in AQP1-null mice and that intracranial pressure was reduced by 56% compared to the wild type, correlating with an enhanced period of survival following brain injuries [114,115]. Similarly, antisense knockdown of a transcription factor that drives AQP1 expression in the choroid plexus reduced AQP1 synthesis and increased the survival of brain edema caused by acute water intoxication [116]. Targeting AQP1 may be a viable therapeutic strategy for other conditions involving CSF production, such as hydrocephalus [117].

AQP4 in ependymal cells and the glia limitans (pericapillary astrocyte processes) functions in exchange and absorption processes of CSF and interstitial fluid (ISF) [118,119]. *aqp4*^−/−^ but not *aqp1*^−/−^ knockout mice showed impaired fluid influx into the ventricular space [120], interpreted as evidence for a predominant role of AQP4. In contrast, Trillo-Contreras and colleagues found that AQP1 and AQP4 had comparable contributions to CSF production and handling, given that *aqp1*^−/−^ and *aqp4*^−/−^ single-knockout mice each showed similar reductions in ventricular volume and interventricular pressure, and that only the double *aqp1* and *4* knockouts significantly altered CSF drainage [121]. These findings support the “Bulat–Klarica–Oreskovic” hypothesis, which proposes that constant CSF formation and distribution in the brain occurs through water exchange between brain capillaries and ISF [122]. The discovery of the glymphatic system [123] provided further evidence for the involvement of AQP4 in CSF homeostasis.

### 2.3. Surface Hydration in the Lungs

AQP1, 3, 4, and 5 have been detected in the airways, including in the alveoli, the tracheal epithelium, and the submucosal glands. Though AQPs have been shown to enable water movement, humidification, and fluid secretion in experimental preparations, studies in *aqp* knockout mice surprisingly failed to support clear roles for them in airway physiology [124,125]. However, it is important to note in genetic deletion models that compensatory changes, including the expression of other AQPs, channels, and transporters, might mask knockout phenotypes in normal conditions and necessitate additional experimental stressors to fully test the consequences of the deficit.

AQP1 and AQP5 are the most abundantly expressed AQPs in the lungs (Figure 1). In the pulmonary endothelium, AQP1 serves as the major path for osmotically driven transcellular water flux across the plasma membranes of the microvascular system [126,127,128]. AQP1 is present on the apical and basolateral membranes of endothelial cells, which form the blood vessel boundaries within the airways and alveolar regions of the lungs [129]. A lack of AQP1 results in a decreased water permeability across the microvascular endothelium [128,130,131], whereas highly elevated pulmonary AQP1 levels are associated with lung injury and fibrosis [127].

In a preclinical asthma model, Dong and colleagues found that AQP1 and AQP5 showed increased expression in both RNA and protein levels in mice after the administration of ambroxol, dexamethasone, and terbutaline, supporting a proposed association between AQP levels and the severity of pulmonary edema [132]. Although some studies have suggested that there is no involvement of AQPs in cystic fibrosis, others link lung AQPs with inflammatory conditions such as asthma, chronic obstructive pulmonary disease (COPD), and acute lung injury [124,125]. The reduced expression of AQP5 in humans results in increased mucus production; genetic polymorphisms in AQP5 have been associated with declines in lung capacity and function in COPD patients [133,134,135]. A recent study suggested that AQPs are also involved in lung development and aging [124].

### 2.4. Secretion of Gastrointestinal Fluids in the Digestive System

Several classes of AQPs are found in the digestive system, including AQPs 1, 3, 4, 5, 8, 7, 9, and 11. In the gastrointestinal tract (GIT), AQPs 1, 3, 4, 5, and 8 are abundant, mediating the secretion of gastrointestinal fluids and facilitating digestive processes [136,137]. AQP1 is largely expressed in the endothelial cells of the stomach, small and large intestine, and the bile duct [138,139,140]. In the GIT, it regulates water transport between the gastrointestinal mucosa and the blood [137,141]. AQP3 is found predominantly in the esophagus as well as the small and large intestines [142,143]. In the colon, abundantly distributed AQP3 dehydrates fecal material by facilitating water absorption [144]. Levels of AQP4 in the stomach correlate with proton transporter expression and influence gastric acid secretion [145]. AQP5 in the stomach and duodenum has been proposed to function in water transport and glandular secretion [146,147]. AQP8 is found in the epithelial cells of the duodenum, jejunum, and colon, and with AQP3 shows reduced expression in inflammatory injury responses [143]. AQPs in the digestive system serve as regulators of water and small solute absorption, serving essential roles in maintaining gastrointestinal fluid balance as well as in the control of cell motility and proliferation.

The mislocalization and aberrant expression of AQPs in the GIT have been associated with the progression of diseases such as gastric and colon cancers, inflammatory bowel disease, gastritis, and diarrhea [57]. The upregulated expression of AQP1 in intestinal blood vessels is associated with gastric cancer progression [148,149]. Increased AQP1 mRNA and protein levels have been detected in tumor biopsies of epithelial gastric adenocarcinoma and correlated with poor prognoses [150]. Bacterial-induced internalization of AQP2 and 3 channels interferes with intestinal water transport and causes a diarrheal phenotype in infected mice [151]. The dysregulation of AQP3 and AQP4 has also been linked with diarrhea [57,152]. AQP3 levels are increased in gastrointestinal tumors; conversely, AQP3 knockdown decreased the invasiveness of gastric cancer cells [153]. AQP5 levels are upregulated in patients with early stage colorectal cancer [154] and facilitate gastric tumor development by augmenting cancer cell invasiveness [155]. Reduced AQP3 and AQP8 expression levels correlate with inflammation and injury in a murine colitis model [156], suggesting potentially protective roles in inflammatory bowel disease. AQPs in the digestive system are important targets for the prevention and treatment of gastric and intestinal diseases.

### 2.5. Glandular Secretions

The average rate of lacrimal production and secretion for humans is estimated to be around 5 mL per day. The transepithelial secretion of lacrimal fluid is mediated by the movement of electrolytes and net water flux down the resultant osmotic gradient [157]. Lacrimal secretion by acinar cells uses molecular mechanisms similar to those involved in pancreatic and salivary secretions, sensitive to second-messenger signaling. AQP4 and AQP5 are present in the apical side of acinar cells and the basolateral side of ductal cells [158]. Mice deficient in *aqp4* were reported to have reduced cAMP-stimulated lacrimal secretion [159]. However, other work reported no detectable difference in lacrimal secretion between pilocarpine-stimulated and basal conditions in mice lacking *aqp1*, *3*, *4*, or *5*, leaving the role of AQPs in physiological secretion in need of further investigation [160].

Human liver hepatocytes produce around 600 mL of primary bile per day that is modified in transit by cholangiocytes, which are specialized epithelial cells lining the biliary tract [161]. Under conditions stimulating bile secretion (e.g., glucagon signaling), AQP8 is relocalized to the canicular plasma membrane in hepatocytes [162,163]. *aqp8* knockout mice had no profound liver dysfunction but were reported to develop mild hyperlipidemia [164], consistent with the hyperlipidemia observed in cholestasis patients with decreased or obstructed bile flow [165]. AQP1 expressed in cholangiocytes is translocated in response to secretin stimulation [166], to regulate ductal bile secretion and water homeostasis [167]. *aqp1* knockout mice had defects in dietary fat processing [168], though technical limitations constrained the measurements of bile secretion to the largest mice in the study, which had the mildest defects. Detailed analyses of the roles of AQP1 in cholangiocyte-dependent bile secretion remain to be done.

## 3. AQPs in Signal Transduction and Sensory Function

### 3.1. Neural Crest Cell Types

Neural crest cells are unique to vertebrates. During embryonic development they generate the neurons and glia of the peripheral nervous system, melanocytes and cranial cartilage, bone, and adipose tissue. Aberrant neural crest cell differentiation causes cancers, developmental syndromes, and birth malformations [169]. AQP1 plays a role in the migration of neural crest cells, with differential AQP1 levels affecting migration speed, direction, and filopodia length [170]. However, the functional role of AQPs in the peripheral nervous system remains incompletely understood. AQP1, AQP2, and AQP4 have all been localized to neurons or glia in trigeminal ganglia, periodontal Ruffini endings, dorsal root ganglia, and the enteric nervous system. Studies in *aqp1*^−/−^ mice suggested that AQP1 in a complex with the voltage-gated sodium ion channel Nav1.8 plays a role in peripheral pain perception [171]. The authors proposed that the inhibition of AQP1 might be a route to achieve analgesia at the presynaptic spinal level, in contrast to earlier studies suggesting that AQP1 has no role in pain perception [172]. These differences remain to be resolved, but based on studies supporting the involvement of AQP1 [173] and AQP4 [174] in pain perception in the central nervous system, Ma and colleagues [175] have suggested that the participation of AQP1 in peripheral pain perception is likely.

### 3.2. Structure and Function in the Eye

AQPs 0, 1, 3, 4, 5, 7, and 11 have been identified in the eye (Figure 1). Of these, the roles of AQPs 0, 1, and 5 are the most extensively studied. AQP0 is thought to be almost exclusively expressed in lens fiber cells (Figure 3B); AQP1 is present in lens epithelial cells; and AQP5 is found in both cell types [176]. AQP0 mutations have been linked with congenital cataracts [177]. The knockdown of AQP1 by siRNA in lens epithelial cells increased the rate of apoptosis [178]. AQP5 dysfunction could promote congenital cataract formation in infants [179]. High levels of reactive oxygen species can lead to cataract formation, providing an interesting link with the H_2_O_2_-transporting capabilities described for the lens AQPs 0, 1, and 5, and highlighting these channels as potential therapeutic targets for treatments of age-related cataract formation [180].

AQP0 has a low water permeability compared to other human AQP paralogs, such as AQPs 1 and 4 (Table 1), but this is mitigated by its high membrane density and organization in lens fiber cells [181], and depends on bilayer composition [182]. The presence of AQP0 enhances the structural integrity and transparency of the lens, facilitating volume regulation, the formation of membrane junctions, nutrient supply, and waste disposal in fiber cells. Structural linkages between cells could result from bridges between AQP0 tetramers and adjoining cell proteins via conserved proline residues in loops A and C, or clusters of positively-charged residues in the AQP0 extracellular loop interacting with negatively charged lipid headgroups in the opposing membrane [183,184,185]. In *aqp0*^−/−^ transgenic mice, the promoter-driven expression of AQP1 in lens fiber cells reduced the severity and slowed the formation of lens cataracts; however, the transgenic lens fiber cells lacked a compact cellular architecture, supporting a structural role for AQP0 in cell–cell adhesion [186].

Ion conductance has been observed for AQP0 reconstituted into lipid bilayers [7,20,187]. However, the physiological role of AQP0 as an ion channel in the eye remains to be established. Cooperativity between AQP0 subunits, the dependence on pH, intracellular Ca^2+^ concentrations, and the phosphorylation status of AQP0 are likely to be important in the regulation of water permeability, maintaining lens clarity [188,189].

In the eye, AQP1 is present in photoreceptors, retinal pigmented epithelia, ciliary epithelial cells, and the trabecular meshwork [190,191,192]. In the ciliary epithelium, AQP1 mediates the secretion of aqueous humor. In the corneal endothelium and stroma, it enables water transport to keep the corneal stroma partially dehydrated, as required for corneal transparency. AQP4 is expressed in the endfeet in Müller glial cells, which span radially across the retina to provide support [193,194]. AQP4 polarization in Müller cells depends on anchoring to α-syntrophin [195] and shows colocalization with the inwardly rectifying K^+^ channel, Kir4.1 [194,196], though K^+^ current amplitudes were not different in Müller cells from *aqp4*-null mice [197].

### 3.3. Hearing and Balance in the Inner Ear

AQPs 1, 7, and 9 are expressed in the inner ear (Figure 1) [198,199,200]; changes in expression are associated with both auditory and balance dysfunctions [201]. In vestibular dark cells and endolymphatic sac epithelial cells, AQPs mediate fluid transport in the inner ear; in sensory and ganglion cells, AQPs contribute to sensory signal transduction [202]. AQP1 in the inner ear mediates auditory function in cochlear cells and balance in vestibular cells (Figure 3B). AQP1 mRNA and protein levels increased when motion sickness was induced in mice, which might be a compensatory mechanism since mice became more sensitive to motion sickness when AQP1 levels were reduced [203]. AQP2 in vestibular cells is regulated by V_2_ vasopressin receptors in the endolymphatic sac that controls the reabsorption of endolymph via a mechanism similar to that in kidney collecting ducts [204,205,206]. Analogous to nephrogenic diabetes insipidus (a failure to concentrate urine), Meniere’s disease, which results in hearing loss and vertigo, is associated with increased expression of the V_2_ receptor [207,208,209,210]. AQP3 in the endolymphatic sac of vestibular cells is responsible for concentrating endolymph [211,212]. AQP4 has a role in potassium homeostasis [213,214], and AQP5 is implicated in low-frequency sound perception [215,216,217]. The function of AQP6 in the cochlear sensory epithelium [218] remains to be defined.

### 3.4. Cardiac Hypertrophy and Edema

Cardiac hypertrophy (the pathological enlargement or thickening of heart muscle) is caused by increases in cardiomyocyte size as well as changes in extracellular matrix (ECM). AQP1, which is expressed in cardiomyocytes, has been suggested as a target for therapeutic intervention based on its role as a H_2_O_2_ channel [49,219,220,221]. It is expressed in both human and mouse cardiomyocytes in hypertrophic conditions, and is co-localized with NADPH oxidase-2 and caveolin-3 [49]. AQP1-mediated flux of H_2_O_2_ was blocked by treatment with bacopaside II, which also attenuated cardiac hypertrophy in an in vivo mouse model [49]. Bacopaside II from the medicinal plant *Bacopa monnieri* was characterized as a blocker of the AQP1 intrasubunit pore in the *Xenopus* oocyte expression system [222].

Cardiac edema in mice after transverse aortic constriction (TAC) resulted in myocardial hypertrophy and cardiac dysfunction compared to control mice. This was associated with increased myocardial water content and AQP1 expression. Treatment with the proposed AQP inhibitor, acetazolamide [223], reduced the ratio of heart weight to body weight, myocardial water content, and AQP1 levels in TAC mice [224,225,226]. Inhibitors of AQP1 could represent new possibilities for treating cardiac edema and hypertrophy.

### 3.5. Skeletal Muscle Viability

In addition to its association with a range of CNS conditions [227], defects in the assembly of the dystrophin-associated protein complex (DAPC) are associated with several muscular conditions. In the CNS, the DAPC anchors AQP4 in astrocyte endfeet membranes [228]. Defects in the glycosylation of α-dystroglycan impair its ability to interact with the ECM and are associated with cobblestone lissencephaly (a set of rare brain disorders associated with a lack of brain folds (gyri) and grooves (sulci), in addition to cognitive impairment). In the context of skeletal muscle, the DAPC has both mechanical stabilizing and signaling roles, mediating interactions between the cytoskeleton, membrane, and ECM. For example, mutations in the gene encoding ε-sarcoglycan, which is a DAPC component, cause myoclonus dystonia, a movement disorder characterized by rapid muscle contractions and/or sustained movements, resulting in abnormal postures [227]. In control muscle, AQP4 did not show a direct interaction with any of the four sarcoglycans, but it co-immunoprecipitated with α1-syntrophin, indicating that this modular protein might link AQP4 levels with the DAPC complex, as seen in the CNS. In six patients with dysferlin deficiency, there was a reduction in AQP4 levels that correlated with the severity of the muscle histopathological lesions, while α1-syntrophin levels were normal. Animal models have demonstrated that the destabilization of the DAPC leads to membrane fragility and loss of membrane integrity, and several studies have implicated the functional coupling of AQP4 and DAPC in muscle disorders. AQP4 can be expressed in muscle despite the absence of the DAPC, and AQP4 can be lost after the onset of muscle degeneration even when the DAPC is present. Overall, it would appear that AQP4 loss in skeletal muscle is associated with muscular dystrophy and its pathogenesis [229,230,231,232,233,234,235].

## 4. AQPs in Defense, Protection, and Support

### 4.1. Blood–Brain Barrier

The blood–brain barrier (BBB) separates the vasculature from the brain (Figure 3C); it is not a simple rigid structure but a rather complex interface between the two. The physical barrier is created by specialized tight junctions between endothelial cells lining the cerebral microvessels, allowing highly selective transport to limit entry to only required nutrients, to prevent the entry of potentially harmful compounds into the brain, and to export toxins and waste products [236,237]. The restriction of molecular, ionic, and fluid exchange at the BBB ensures a protected brain microenvironment for neural signaling [119] and guards against the effects of systemic fluctuations after a meal, exercise, or other factors [238].

In addition to the endothelial cells, the BBB is augmented by pericytes and astrocytes. The location of astrocytes between the capillaries and neurons allows them to mediate intercellular communication. The astrocytic perivascular endfeet near the vascular interface have an essential role in regulating water homeostasis for the whole brain through highly expressed AQP4 (Figure 3C). The revolutionary discovery of the glymphatic system in 2012 explained for the first time how waste products could be removed from the brain [123], preventing the accumulation of metabolic waste that can lead to neurodegeneration. The function of the glymphatic system is dependent upon astrocytic AQP4 [239], and shows enhanced activity during sleep [240]. AQP4 appears to have a critical role in Alzheimer’s disease through the proposed clearance of amyloid-β [241]. AQP4 is also found in peri-synaptic sites, where it is thought to assist with neurotransmitter clearance as part of the tripartite synapse [242]. Astrocyte processes in contact with non-myelinated axons and nodes of Ranvier are proposed to enable the clearance of potassium ions [243].

AQP4 is directly associated with brain edema formation under pathological conditions such as ischemia and trauma. Conditional knockout of *aqp4* selectively in glia showed a 31% decrease in blood–brain water uptake in response to hypo-osmotic treatment [244]; conversely, glial-specific overexpression of AQP4 accelerated water uptake [245]. A special feature of AQP4 at perivascular endfeet is the presentation of polarized and highly dense structures known as orthogonal arrays of particles (OAPs) [246]. AQP4 levels increased at the astrocyte cell surface despite no changes in total protein expression [31,247], suggesting that relocalization of the channel occurred after traumatic brain injury (TBI), infections, tumor growth, or stroke [248], which could exacerbate brain swelling [249,250]. CNS injuries associated with a disrupted oxygen supply (hypoxia) have been identified as triggers for rapid sub-cellular relocalization of AQP4 from intracellular pools to the cell surface [31]. In stroke models [251] and TBI [252] *aqp4*^−/−^ mice did not exhibit elevated brain water content, confirming the connection between cerebral edema and AQP4. Treatment with trifluoperazine following spinal cord injury in rats provided a proof of concept that preventing AQP4 sub-cellular relocalization is a viable strategy for treating cytotoxic edema [31]. Even though high AQP4 surface expression is implicated in the formation of cytotoxic edema in the early post-injury stage when BBB integrity has not been compromised, it also contributes to water clearance in the later stage of vasogenic edema [253,254]. Th modulation of AQP4 localization adapted to the stage of edema holds promise as a therapeutic strategy.

It has been noted that AQP4 is important for BBB integrity, and that BBB integrity influences AQP4 expression [255]. Structural molecules, such as agrin and laminin, are necessary for BBB integrity [256,257] and orchestrate AQP4 polarization within glial cells. Culturing primary mice astrocytes with the neuronal agrin paralog A4B8 led to increased membrane clustering of AQP4 in OAPs, with increased levels of the M23 splice variant [258]. The same effect was observed in primary rat astrocytes in the presence of laminin [259]. Mice lacking endothelial agrin showed downregulated AQP4 surface levels [260]. Finally, the genetic deletion of dystrophin [261] or α-syntrophin [262,263] was accompanied by a loss of AQP4 polarization in mice and resulted in protective effects against edema formation similar to those in the *aqp4* knockout mouse, demonstrating that membrane tethering and polarization within astrocytes is important for AQP4 function. BBB cells from mice lacking α-dystrophin, grown in two- and three-dimensional co-culture systems, showed leakier barrier function, scattered AQP4 localization, and reduced AQP4 expression levels compared to wild-type controls [264], reflecting both the role of the DAPC in AQP4 anchoring and the influence of AQP4 in maintaining the BBB.

The inwardly rectifying potassium channel Kir4.1 is associated with the DAPC through α-syntrophin binding [265]. AQP4 and Kir4.1 are co-localized at astrocytic endfeet, though studies have not found evidence for functional coupling [266]. However, increased brain levels of K^+^ in epilepsy patients [267] were seen with elevated AQP4 levels [268] and with decreased dystrophin in rats [269]. A loss of polarization in Alzheimer’s disease could impair solute clearance and increase the aggregation of β-amyloid [270].

AQP4 is essential for astrocyte plasticity [271]. After neuronal damage, reactive astrocytes undergo AQP4-dependent migration to lesion sites and isolate dying neurons by forming glial scars to limit the spread of tissue damage. Astrocytes from wild-type mice showed a three-fold higher migration rate and 50% faster wound healing rate in comparison to those from *aqp4*^−/−^ mice [254]. Glial scars, while beneficial in the short term, appear to be counterproductive to long-term functional recovery [272]. AQP4 in activated microglia in glial scars indicates a possible role for this channel in microglial migration as well [273].

### 4.2. Skin Hydration and Wound Healing

Skin is an organ which provides a physical, thermal, and microbiological barrier, and regulates water loss via evaporation. Six AQPs (AQPs 1, 3, 5, 7, 9, and 10) are expressed in various human skin cell types [274]. Physiological roles thus far have been attributed to the aquaglyceroporin AQP3 in the epidermis (Figure 3C) and AQP5 in the sweat glands. AQP3 is abundantly expressed in the terminally differentiated keratinocytes in mammals [275] that form the stratum corneum (SC), the most superficial skin layer of the epidermis which is essential for skin plasticity, hydration, and barrier integrity. AQP5 expression in sweat glands was confirmed in humans and rodents [276,277]; however, *aqp5* knockout mice did not show overt impairment of sweat gland secretion [277].

Skin hydration is crucial for integrity and barrier function. Changes in AQP3 levels have been linked to various skin diseases, for example, involving the hyperproliferation of keratinocytes, leading to atopic eczema [278], psoriasis [279], or skin cancer [280]. Conversely, the loss of AQP3 has been linked to epidermal spongiosis [274]. Although *aqp3*^−/−^-null mice had no alteration in skin structure [281] or lipid–protein profile [282], they did show defective SC hydration, dry skin, and a two-fold reduction in skin conductance [281]. Elevated humidity did not correct the deficit. Epidermal glycerol and water content in the SC was low, and the secretion of water and glycerol was reduced four- and two-fold as compared with wild-type controls, with a concomitant reduction in skin elasticity [282]. Barrier properties remained intact; however, the time required to restore barrier function after compromise was doubled. Subsequent work demonstrated the beneficial effects of glycerol application for restoring hydration, elasticity, and barrier function [283], suggesting that glycerol is more crucial than water for skin hydration. Glycerol transport also supports lipid synthesis for skin cell proliferation. However, for re-epithelialization during wound healing, AQP3 channels are required to facilitate cell migration from surrounding epidermis, as shown in human keratinocyte cultures [284]. Additional roles of AQPs in skin health and barrier function remain to be defined.

### 4.3. Vascular Endothelial Function and Angiogenesis

The vascular system, one of the first developing organs in an embryo, is formed through vasculogenesis, a process in which new blood vessels are created from angioblasts that differentiate into endothelial cells [285,286]. Networks of blood vessels must be established for the subsequent development of other organs, and serve as scaffolds for angiogenesis throughout life [287]. The endothelium is not a passive barrier, but rather contributes to vascular homeostasis by facilitating the flow of nutrients, signaling molecules, and blood cells. In angiogenesis, new vessels arise from existing vascular networks via the proliferation and migration of endothelial cells in response to the metabolic needs of developing or healing tissues [287]. Angiogenesis peaks during embryogenesis, while in mature blood vessels endothelial cells show slow turnover rates [288]. The endothelial cell layer enables dynamic exchange between the blood and interstitial fluids, influencing vascular tone and regulating vascular permeability. The endothelium regulates inflammatory and proliferative processes and maintains an anti-thrombotic microenvironment [289,290]. Cardiovascular risks associated with endothelial dysfunction reflect imbalances in the signals for vasodilation, anti-thrombosis, inflammation, and proliferation, and can result in pathologic conditions characterized by vascular lesions, endothelial hyperpermeability, morphological atherosclerotic changes, and excessive angiogenesis [290,291,292].

AQP1 and AQP3 are the most prevalent AQPs of the endothelium [293]. Although AQPs have not been found in cerebral endothelia [294], AQP1 is abundant throughout the peripheral vascular endothelium supplying multiple organs such as the kidney [295], lung [129], digestive tract [296], cardiovascular system [297], skeletal muscle [297], and the eye [298]. AQP1 in endothelia is implicated in the control of vascular permeability, the formation of blood vessels by angiogenesis [299,300,301], endothelial cell migration, and wound healing [302,303,304]. The absence of AQP1 has been associated with endothelial dysfunction and atherosclerotic progression [293,305,306].

Endothelial cell expression of AQP1 in response to hypoxia and inflammation is regulated by signaling, transcription factors, and epigenetic modifications. AQP1-mediated fluxes of nitric oxide (NO) in endothelial cells could contribute to endothelium-dependent relaxation [70,307,308,309], setting vascular tone [310], and protecting vessel walls from inflammation and thrombosis. NO decreases platelet adhesion and aggregation, and reduces endothelial migration and proliferation, sustaining a quiescent state [290,291]. In inflamed tissues with low AQP1 levels, NO availability is reduced and vasorelaxation is impaired [291,306]. The flow-dependent transcription factor Krüppel-like factor 2 (KLF2) increases AQP1 and endothelial NO synthase (eNOS) gene expression, ensuring NO availability and promoting an athero-protected non-inflamed endothelial state [306].

The regulation of AQP1 expression levels by hormonal signaling impacts cell migration and angiogenesis. The growth factor erythropoietin (EPO) in the presence of extracellular Ca^2+^ elevates AQP1 in endothelial cells [302], inducing angiogenesis, neovascularization, and migration, which is impaired by AQP1 siRNA silencing [302,311,312,313]. Estrogen also increases AQP1 expression levels, leading to vascular tubulogenesis; in contrast, downregulated AQP1 decreases tube formation in human umbilical vein endothelial cells [314]. Genes encoding eNOS and vascular endothelial growth factor (VEGF) exhibit DNA methylation sites in their promoter regions, supporting a role for epigenetic pathways in governing endothelial function and AQP1 levels [293,315,316].

Pathological environments are often associated with hypoxia, pH changes, increased levels of ROS and oxidative stress promoting NO imbalance, cytokine activation, impaired vessel integrity, and infiltration of inflammatory molecules [317,318,319], which highlight AQP1 as a key factor. Genetic variants of *aqp1* have been found to be over-represented in heritable pulmonary arterial hypertension patients [320]. Hypoxia drives Ca^2+^-dependent increases in AQP1 protein levels, enhancing the migration of rat pulmonary artery smooth muscle cells, whereas AQP1 silencing impairs hypoxia-induced migration [321]. Risk factors such as pulmonary hypertension have been correlated with increased AQP1 levels [321,322,323]; conversely, cases of established pulmonary hypertension could be restored by AQP1 silencing in a mouse model [323]. AQP1 in atherosclerotic lesions and malignant tumors has been suggested to promote plaque progression and angiogenesis [305,324]. Although the specific molecular mechanisms of AQP1 in vascular physiology and pathology are not fully understood, AQP1 channels appear to serve important protective roles in the cardiovascular system [293,304,305,306]. The roles of AQP1 in vascular and cardiac disorders are a promising area of current exploration for new therapeutic interventions.

### 4.4. Inflammatory and Immune Responses

AQPs are involved in immune cell priming by inflammatory triggers, migration, and pathogen elimination in inflammation and immune system responses [87,325]. AQP1 and AQP9 levels were elevated in human leukocytes after intravenous or in vitro lipopolysaccharide (LPS) stimulation [326,327]. LPS stimulation increased AQP9 expression in murine dendritic cells (DCs), human macrophages, and neutrophils [328]. AQP1, 3, and 5 are associated with the activation and proliferation of activated T lymphocytes; similarly, AQP3 and 5 are regulated in immature DCs [325]. NLRP3 activation of macrophages and AQP-induced cell swelling [329,330,331] lead to IL-1β and IL-18 release [332].

Morphological changes required in immune cell responses include endocytosis (for antigen presentation), chemotaxis-induced migration to reach infection sites, and phagocytosis to fight invading pathogens, all of which are influenced by AQPs. Antigen uptake by DCs and migration to the lymph node have been suggested to require AQP5 [333] and AQP7 [334]. Migration has been linked to AQP1 in macrophages [335], AQP3 in T-cells, and AQP5 and 9 in neutrophils [336,337]. Chemokine-dependent T-cell migration seems to require AQP3-mediated H_2_O_2_ uptake [338]. The inhibition of AQP4 with AER-271 in mice T-cells reduced the expression of key chemokine receptors (S1PR1, CCR7) and impaired chemotactic activation [339]; however, evidence for this agent as a selective AQP4 inhibitor is equivocal, and the effects could reflect the inhibition of IKKβ and NF-κB signaling [340]. Phagocytosis, an ultimate defense mechanism, is supported by AQP3-mediated glycerol flux in macrophages during phagosome formation [341]. It is no surprise that AQPs are linked to a large number of inflammatory diseases in various tissues, such as asthma, acute kidney injury, diarrhea, endocrine pancreas dysfunction, and osteoarthritis [342].

### 4.5. Physical Membrane Compliance

Specialized subsets of cells in the human body are required to accommodate rapid volume changes imposed by pressure or mechanical strain via adaptations enabled by AQP1 and AQP4 channels. Glaucoma is characterized by increased intraocular pressure, often caused by obstructed fluid movement through trabecular meshwork, which forms part of the conventional outflow pathway for aqueous humor in the eye. Patients with primary open-angle glaucoma showed elevated endothelin-1 levels, a suppressor of AQP1 transcription [343]. In trabecular meshwork, AQP1 levels were manipulated using adenovirus-mediated overexpression and antisense knockdown to assess function [344]. AQP1 knockdown yielded a small decrease in trabecular meshwork resting volume (~8%) with consequent increases in paracellular outflow, whereas overexpression led to the opposite outcome; subsequent work showed that the impact on overall outflow dynamics was minimal [345], suggesting another function for AQP1. A novel role for AQP1 in cell compliance (i.e., allowing rapid volume changes under mechanical strain) was demonstrated in primary human trabecular meshwork cells that when exposed to lateral stretch showed an upregulation of AQP1 at levels correlated directly with tension force [346]. Cytotoxicity was inversely proportional to AQP1 level, confirming the protective role of AQP1 in trabecular meshwork during mechanical stress.

AQP4, localized in the sarcolemma of skeletal muscle, similarly has been proposed to have a protective role in reducing damage to cell membranes during the volume changes that are imposed by muscle contraction [229]. During pathogenesis, AQP4 is lost as muscle degeneration progresses in dystrophies linked to mutations in dystrophin and sarcoglycan genes [229]. Continuing work in the field is likely to identify other examples of AQPs that enhance physical compliance as well as benefit cell survival and function during pressure and volume stress.

### 4.6. Transport of Nutrients

Metabolic functions in the human body require glycerol transport facilitated by aquaglyceroporins in adipocytes, the liver, and the small intestine [110,347]. Triglyceride fatty acid cycling between adipose tissue and liver is regulated by nutritional state [348], and glycerol as an energy source is involved in the balance between lipid accumulation and hepatic gluconeogenesis. Glycerol is normally stored as triacylglycerol (TAG) in adipocytes, but during fasting TAG is hydrolyzed into fatty acids and glycerol, released into blood, and taken up in the liver [349,350]. Liver glycerol is phosphorylated into glycerol-3-phosphate as a precursor for gluconeogenesis and TAG synthesis [351].

Human adipocytes express AQPs 3, 7, 9, 10, and 11 [89,90,352]. AQPs 7 and 10 are proposed to enable glycerol efflux from adipocytes [350]. Lipolytic stimuli (such as isoproterenol and leptin, simulating a meal intake) increase the membrane abundance of AQP7 and AQP10, whereas lipogenic stimuli (such as insulin or fasting) cause negative regulation [89,352]. AQP9 levels in plasma membranes were not affected by either treatment [352]. Studies with *aqp7*^−/−^-null mice showed a three-fold reduction in glycerol release into the bloodstream [353,354], leading to adult-onset obesity and insulin resistance; the knockdown of AQP7 in mouse 3T3-L1 adipocytes resulted in two-fold lower glycerol secretion [354]. However, conflicting studies failed to reproduce the results [355,356]. In contrast to the effects observed in rodents, a lack of AQP7 in humans did not correlate with obesity or type two diabetes [357,358]. A loss-of-function mutation in human AQP7 did not associate with any overt metabolic phenotype, although exercise-induced increases in plasma glycerol were abolished [359]. Differences between species were also seen for the *aqp7* promoter, in which a common polymorphism (A-953G SNP) correlated with obesity in mice but showed no effect in humans [360]. Subsequent studies identified AQP10 as a pathway for glycerol efflux from human adipocytes [89]. AQP10 is not expressed in mice [361], supporting the stronger reliance on AQP7 for glycerol transport in rodents as compared to humans.

Once glycerol is released into the blood it can be taken up by AQP9 into the liver to be used for gluconeogenesis [351], and also serves as an energy supply for heart and skeletal muscle [347,362]. In the liver, AQP8 is suggested to play a role in bile formation [363], and AQP9 presents the main entry route for glycerol. Liver glycerol permeability in mice correlated with AQP9 membrane abundance and was absent in *aqp9* knockout mice [364]. Gender-linked differences in AQP9 levels in the liver are consistent with gender dimorphism in metabolism [365,366]. Insulin downregulated hepatic AQP9 mRNA levels in mice by interacting with the negative insulin response element (IRE) in the promoter region of the *aqp9* gene [367]. Effects remain to be confirmed in humans.

Consistent results for AQP3, AQP7, and AQP9 expression are lacking in humans with obesity, insulin resistance, and type two diabetes [347], and the physiological roles of AQPs 3 and 7 in the liver are uncertain [352]. Abnormally high levels of adipose AQP7 and liver AQP9 were detected in obese insulin-resistant *db*^+^/*db*^+^ mice in combination with elevated glycerol and glucose, worsening hyperglycemia in the diabetic state [367]. The upregulation of AQPs 3 and 9 after insulin treatment and downregulation of AQP7 and AQP9 in response to leptin have been observed [368]. Glycerol metabolism in humans, the involvement of AQPs, and differences between species in various metabolic diseases require further investigation.

The brain has a high demand for energy, accounting for at least 20% of the body’s energy consumption [242]. As AQP9 transports lactate [85], it is speculated to be involved in neuronal energy metabolism via the astrocyte-to-neuron lactate shuttle (ANLS) [369,370]. AQP9 mRNA and protein have been confirmed in rodent brains [84,371,372], but the relevance to human brain is unclear. An antibody validation study reported no AQP9 protein expression in the human cerebral cortex, hippocampus, or cerebellum [373], and minimal levels of AQP9 RNA in human cortical astrocytes [374]. In rats, a short AQP9 isoform is enriched in astrocyte mitochondria, a subpopulation of dopaminergic neurons [371,372] and retinal ganglion cells (RGCs) [375].

According to the ANLS theory, astrocytes are thought to take up glucose at the BBB, create lactate, and release it to neurons. Lactate can be transformed by neurons into pyruvate, used for ATP production, and in some cases appears to be preferred over glucose as an energy substrate [375]. AQP9 expression improves RGC survival under stress conditions [376,377]. A higher survival rate in wild-type mice RGCs as compared to those from *aqp9*^−/−^ mice correlated with an approximately 15% higher concentration of intraretinal lactate [375]. *aqp9* deletion alone did not lead to RGC death, raising the possibility of other routes for lactate uptake via monocarboxylate transporters [375]. Increases of 50% in glucose levels were detected in *aqp9*-null RGCs, suggesting a compensatory mechanism for energy supply. AQP9 facilitates glycerol uptake into neurons after ischemia [378,379] and regulates cell volume in hypoxia-exposed RGCs [380]. AQP9 in RGCs has a role in functional maintenance by facilitating lactate influx and regulating cell volume under stress.

### 4.7. Detoxification

AQP9 carries a broad range of substrates, including urea, monocarboxylates (lactate, β-hydroxybutyrate), polyols (glycerol, mannitol, and sorbitol), purines, and pyrimidines [83]. Fluxes of arsenite [80,381], ammonia, and urea [382] were surprising additions to the group of permeable substrates, pointing to a role for AQP9 in detoxification by moving these molecules from blood into the liver for hepatic clearance [80]. Human arsenic exposure occurs through poisoning from contaminated food and water as well as from treatments for some cancers, such as promyelocytic leukemia [383]. Arsenic trioxide, a metabolite of arsenic, shows fluxes through rat and human AQPs 7 and 9, but not AQPs 3 or 10, expressed in *Xenopus* oocytes [80]. The survival of arsenic toxicity in mice was reduced in *aqp9*-null mice compared to the wild type [381].

Neurotoxic ammonia produced by amino acid metabolism can cross the BBB and needs to be removed to avoid hyperammonemia. Clearance is dependent on AQPs 8 and 9. The main pathways for elimination in humans are hepatic ureagenesis and glutamine synthesis. AQP8, predominantly localized in the inner mitochondrial membrane, efficiently transports ammonia in expression systems [382,384,385,386,387]. AQP8 knockdown by siRNA in primary rat hepatocytes decreased basal ureagenesis by about 30% in increased ammonia conditions [388]. The stimulation of ureagenesis by glucagon increased AQP8 protein expression and ammonia permeability. In hypothyroidism, boosted hepatocyte urea synthesis was correlated with elevated AQP8 expression [389]. Urea permeability was also demonstrated for rat AQP9 channels expressed in *Xenopus* oocytes [390] and reconstituted in proteoliposomes [391]. Following ureagenesis, urea efflux is mediated by AQP9. Reduced urea permeability was observed in *aqp9*-null mouse hepatocytes and in wild-type hepatocytes treated with the AQP9 inhibitor phloretin. These channels are likely to work in combination with separate AQP-independent pathways involving urea-transporter-like proteins [392].

## 5. AQPs in Cell Motility and Cancer

### 5.1. Mechanisms of Cell Migration

Cell migration, a tightly regulated mechanism, is crucial for various physiological processes, including morphogenesis, immune responses, and wound healing. Under pathological conditions, cell migration favors tumor growth and progression by enabling the expansion of the vascular endothelium into new blood vessels supplying tumor metabolic demand, pre-metastatic niche formation, and cancer cell dispersal into distant tissues [393,394,395].

AQPs play a major role in migration (Figure 3D) as well as proliferation, tumor growth, angiogenesis, and invasion [396,397,398,399,400]. Correlations between AQP levels and cancer malignancy have prompted hypotheses that AQPs might be useful as prognostic indicators or therapeutic targets [68,401,402,403,404]. Interestingly, AQPs upregulated in some tumors are not found in the tissue of origin, supporting the idea that aberrant expression is a pathological hallmark [405]. Augmented AQP expression levels at the leading edges of migrating cells might speed local volume changes to facilitate cytoskeletal and ECM modifications required for locomotion [399,406,407]. Various classes, including AQP1, AQP4, AQP5, and AQP9, have been found to polarize at the leading edges of migrating cells depending on cell type [67,336,408,409,410], and might contribute to all stages of movement, with roles in polarization, protrusion, adhesion, and retraction. A single class of AQPs can support migration across diverse cell types; AQP1 for example is linked to motility in glia, epithelia, microvascular endothelia, and other cells [406,407,408,411,412].

In response to chemotactic stimuli, migrating cells undergo polarization along the primary axis of movement, seen in both expression patterns and signaling events [413]. During polarization, AQP1 activity in concert with other channels and transporters is thought to support lamellipodial formation by clearing space for actin polymerization [399,414], resulting in membrane extension and protrusion [396,415]. During locomotion, cells must cyclically detach and re-adhere to the ECM, orchestrated by a dynamic focal adhesion turnover process [416]. Cell–ECM interactions depend on the integrins connecting the ECM and intracellular actin filaments [417,418]. Several AQPs interact with ECM-associated adhesion molecules, migration-related regulators, and matrix metalloproteinases (MMPs) [419,420,421,422]. In mesenchymal stem cells from bone marrow, elevated AQP1 correlates with increased β-catenin and focal adhesion kinase, key regulators in migration [419,423,424]. The last stage, retraction, is characterized by Ca^2+^ influx and K^+^ efflux. A proposed mechanism suggests that water intake during lamellipodial formation stretches the plasma membrane [302], accruing tension that opens stretch-activated Ca^2+^ channels and causes the detachment of adhesion proteins at the trailing edge [425], while parallel K^+^ loss could enhance AQP-mediated water efflux, decreasing cell volume and contracting the rear of the cell [426].

AQP1-mediated migration is involved in both physiological and pathological conditions. AQP1 water transport supports axonal outgrowth and the regeneration of dorsal root ganglion cells; AQP1 deficiency causes impaired axonal growth [427], which can be rescued by transfection with wild-type AQP1 and AQP4, but not by an AQP1 mutant deficient in water permeability [427]. In the proximal tubule, AQP1 enhances the migration of kidney epithelial cells [408]. In pathological conditions, AQP1 is required for the MMP2- and MMP9-dependent migration of lung cancer cells in vitro [420]. Similarly, in human gastric carcinoma cells, upregulated AQP3 levels correlated with increased MMP2 and MMP9 expression, whereas AQP3 silencing reduced MMP2 and MMP9 expression [428]. The exact mechanisms by which AQPs influence the expression of MMPs to enhance ECM degradation remain elusive, but could involve interactions with integrins. AQP1 enhanced the migration of neuroblastoma cells under hypoxic conditions [412]. AQP1 but not AQP4 accelerated migration in glioma cell lines [411], suggesting that more than water permeability alone was needed for AQP1-facilitated migration. The cation channel activity of AQP1 enhanced the migration of colon cancer cells, an effect blocked by the AQP1 ion channel inhibitors AqB007, AqB011, and 5-hydoxymethyl furfural [429,430]. AQPs promote the cell migration of a variety of cell types in both physiological and pathological processes, including angiogenesis, neuronal and epithelial regeneration, as well as tumor metastasis. With the development of specific modulators, AQPs could constitute promising therapeutic targets for selective intervention in these events.

### 5.2. Cancer Invasion and Metastasis

Cancer has a high incidence worldwide and is a leading cause of death [431]. Cancer treatments comprising surgery, chemotherapy, and radiation are mainly focused on cancer cell proliferation and are limited by serious side effects [432,433]. Although current therapies can provide life-prolonging or even curative effects, highly aggressive and invasive forms of cancer, such as lung cancer, prostate cancer, glioblastoma, breast cancer, and colorectal cancer, persist as major clinical challenges [434,435,436,437,438]. The majority of cancer deaths are caused by the metastasis of the primary tumor to secondary tissue locations [439]. Investigating the mechanisms of cancer invasion and designing tools targeted at cancer cell metastasis in combination with current therapies could open up new possibilities to reduce mortality for highly invasive cancers [440,441].

Among the thirteen higher mammalian AQPs, AQP1 and AQP9 have been associated thus far with stages of the metastatic cascade, including angiogenesis [407], cell migration, cell–matrix interactions, epithelial–mesenchymal transition (EMT), invasion [442], and increased ROS levels associated with oxidative stress and tumor formation [300,405,443]. Research has focused mainly on AQPs 1, 3, 5, and 9 in the progression of various cancer types.

AQP1 overexpression has been shown to increase cellular migration and enhance extravasation from blood vessels and the invasion of tissues [399], while the loss of AQP1 functionality reduces angiogenesis, tissue invasion, and metastasis in many cancer types, including glioma [411], melanoma [444], mammary gland [445], colon [446], and lung cancers [398,420,447]. AQP1 silencing is associated with enhanced levels of the tumor suppressor p53 in pulmonary artery smooth muscle, while the hypoxia-induced upregulation of AQP1 decreases p53 levels and increases proliferation [323]. AQP2 has been linked to estrogen-induced migration, invasion, and adhesion in endometrial adenocarcinoma cells [448]. AQP3 found in multiple carcinomas, such as breast [449,450], prostate [451], gastric [452], colorectal [453], and lung [454] cancers, promotes invasion, metastasis, and EMT. AQP4 upregulation in human edematous brain tumors, such as astrocytomas and adenocarcinomas, correlates with a metastasizing phenotype [455], and in malignant glial cells enhances infiltration and invasion [421]. Increased AQP5 levels have been associated with invasion and metastasis in various tumor types, such as breast [456], ovarian [457], prostate [458], colon [459], liver [460], and lung [461] cancers. AQP5 silencing results in the reduced proliferation and migration of human epithelial breast [456], ovarian [457], and prostate [458] cancer cells, and reduced AQP5 levels impair EMT in colorectal [462] and metastatic liver cancer cells [463]. AQP5 overexpression in lung cancer has been linked to the activation of epidermal growth factor receptor [464], cell invasion, detachment of cell–cell contacts, and loss of cell polarity [465]. AQP6 has been reported to be upregulated in benign ovarian tumors [466]. AQP7 is suggested to be involved in metabolic regulation in breast cancer [467] and has been reported to be upregulated in thyroid cancer [403]. In contrast to the AQPs mentioned above, which are mostly overexpressed in multiple cancers, AQP8 has been found to be reduced in colorectal tumors, while healthy colon tissues show higher AQP8 expression levels [468]. In line with these findings, Wu and colleagues reported that AQP8 overexpression leads to reduced proliferation and aggressiveness of colorectal cancer cells in vitro, impaired tumor growth and metastasis, and higher efficacy of chemotherapeutic drugs in vivo [469]. AQP9 is elevated in astrocytic brain tumors [470,471], prostate [422], breast, colorectal, and gastric cancers, and decreased in lung cancer [472] and hepatocellular carcinoma cells [473]. Increased AQP9 levels in colorectal tumors have also been reported to promote sensitivity to chemotherapy [474]. In hepatocellular carcinoma cells, AQP9 overexpression, accompanied by reduced Wnt/β-catenin signaling and EMT-related molecules, inhibited proliferation, migration, and invasion [473]. Conversely, in renal cell carcinoma, increased AQP9 levels have been linked to aggressive progression and poor survival [475].

Different AQPs have profound impacts on tumorigenesis and metastasis, depending on tumor type, and confer a case-specific matrix of changes in vascular permeabilities, fluid transport, motility, and signaling, comprising a powerful but complex array of targets with prognostic and therapeutic potential [440].

### 5.3. Tumor Angiogenesis

The proliferation of aggressive cancer cells in dense tumors results in hypoxia and oxidative stress factors that drive angiogenic responses, building new blood vessels to supply oxygen and nutrients, provide gas exchange, and remove waste products [476]. Angiogenesis ensures normal tissue health, but in cancers counterproductively abets tumor survival and provides paths for tumor cell spread to distant sites [476,477,478]. High levels of angiogenic factors associated with aggressive cancers drive the reorganization of the tissue microenvironment and induce ECM breakdown [479]. Activator and inhibitor proteins in addition to chemical signals released from tumor cells into the surrounding tumor stroma [480,481] include transcription factors (such as hypoxia-inducible factor 1-alpha (HIF-1α)), growth factors (such as VEGF, epidermal growth factor, platelet-derived growth factor), cytokines (such as TNF, IL-8), and proteases (such as cathepsin and metalloproteinases) [479,482]. Under hypoxic conditions, the transcription factor HIF-1α binds to diverse genes involved in angiogenesis and proliferation, including the HIF-1α-mediated expression of VEGF in addition to concomitant endothelial cell migration and proliferation [483,484].

AQP1 expression and activation in vascular endothelial cells is sensitive to multiple angiogenic factors [294,300,443,485]. For example, the *aqp1* gene promoter contains a consensus HIF-1α binding motif [486]; AQP1 mRNA and protein levels are elevated under conditions of low oxygen [322,412,487,488], supporting the presumption that AQP1 activity can be regulated by the interplay of growth factors within the tumor microenvironment [489,490]. Secondly, it has been shown that AQP1 knockdown in breast adenoma mice models resulted in decreased tumor formation and microvascular density within the tumor as well as reduced lung metastasis [447]. Furthermore, *aqp1* gene disruption in mice with tumor cell implantations [407] and siRNA silencing of AQP1 in the chick embryo chorioallantoic membrane [491] impaired angiogenesis. Inhibition of AQP1 using the bumetanide-derived small molecule AqB013, shown to block AQP1 and AQP4 water channel function in *Xenopus laevis* oocytes, impaired angiogenesis in in vitro model systems [492]. Finally, upregulated AQP1 levels have been found in various vascularizing and metastasizing cancer types, including brain [485], breast [493], lung [494], and colorectal [495] cancers, and are associated with a poor prognosis, especially in tumor stages with increased hypoxic conditions [398,412,496,497].

Numerous studies have linked AQP expression and function to several tumor types and grades, even in cell types that do not usually express water channels. AQPs constitute an attractive target in cancer diagnosis and therapy [67,405].

## 6. Physiological and Pharmacological Modulation of AQP Channel Activity

### 6.1. Intracellular Signals Regulate AQP Expression and Function

Diffusible signaling molecules such as cAMP, cGMP, Ca^2+^, and downstream protein kinases mediate the complex regulation of AQP channel activities, controlling adaptive increases or decreases in functional transport properties and adjusting levels of expression [1]. cAMP and cGMP are involved in numerous biological mechanisms, such as cell growth, differentiation and adhesion, hormone and neuronal signaling, as well as in the regulation of protein kinases, phosphodiesterases, and cyclic-nucleotide-gated ion channels [498,499,500,501,502,503]. Upon ligand binding, the enzyme guanylyl cyclase catalyzes the formation of cGMP from its precursor guanosine triphosphate. Dependent on the localization of guanylyl cyclase, it can be stimulated by peptides at membrane-bound receptors or activated by NO in a soluble cytoplasmic form, linked to the induction of crucial signaling pathways. Not surprisingly, impaired cGMP signaling is associated with pathophysiological outcomes, including cancer and neurological disorders [504,505,506,507]. Intracellular free Ca^2+^ is a potent second messenger needed for a diverse array of cellular responses, including neurotransmitter and hormone release, muscle contraction, as well as the regulation of diverse types of ion channels that sculpt the form and frequency of excitable membrane events conducted as action potentials that carry information in nerves, muscles, and sensory as well as secretory cells [508].

Providing capacity for rapid modifications of membrane physiology by governing channel function, post-translational mechanisms include direct binding and phosphorylation [1]. Cyclic nucleotides can modulate AQPs by direct interaction with a gating domain or by the activation of associated protein kinases, in turn altering the AQP channel via phosphorylation. For example, AQP1 directly binds cGMP at an intracellular gating domain (loop D) to activate cation conductance [23,42]. AQP2 C-terminal phosphorylation by PKA shifts the binding affinity for interacting intracellular heat shock proteins [509]. Ca^2+^ binding regulates channel function via CaM in AQP0 [510,511]. In AQP1, a sequence reminiscent of an EF-hand motif associated with Ca^2+^ binding was suggested, but its functional role in the C-terminal domain remains to be demonstrated [512]. Tyrosine-kinase-mediated phosphorylation modulates the ion channel activity of BIB, which serves a neurogenic role in *Drosophila melanogaster* nervous system development [25,513,514,515]. Other post-translational modifications, including glycosylation, ubiquitination, and lipid modifications, can impact stability, location, and turnover [516,517,518,519], with long-term modulatory roles in channel functions and protein–ligand interactions.

Key consensus sequences for critical sites of modulation are conserved in AQPs across species [520]. Pore properties can be differentially regulated within single types of AQPs, illustrating a powerful capacity for fine-tuning the contributions of multifunctional AQP channels. For example, gating mechanisms distinguish between AQP10 water channel and glycerol channel fluxes through the same pore [521]. Gotfryd and colleagues identified a highly conserved histidine residue (H80 in human AQP10) located in the cytoplasmic vestibule of the intrasubunit pore as critical for the regulation of permeant substrates. At low pH, protonated H80 reorients to interact with a highly conserved glutamate (E27) in the first transmembrane domain, triggering propagated structural rearrangements that result in pore widening, accommodating glycerol permeation without altering pH-independent water permeation. Similarly, intrinsic molecular gates in AQPs differentially regulate water channel versus ion channel activities in the subset of AQPs that can function as dual water and ion channels. The cytoplasmic loop D spans between the fourth and fifth transmembrane domain in each subunit, carrying amino acid sequences that are highly conserved within a given class of AQPs across phyla but not required for osmotic water permeability, suggesting another critical role. Molecular dynamic simulations suggested the binding of cGMP to a poly-arginine motif in human AQP1 loop D, which triggered a conformational rearrangement that allowed the hydration of the central pore of the tetramer and cation permeation [42]. The model was verified using site-directed mutagenesis and voltage clamp recordings, showing that alterations in key residues in the conserved loop D sequence altered the gating of the ion channel in AQP1 without disrupting water flux [42,43]. Mutations in hydrophobic barrier residues lining the central pore altered the ionic selectivity of the cGMP-activated conductance [24].

Enabling an array of long-term adaptations on a time scale of hours to days, transcriptional mechanisms alter the rates of messenger RNA synthesis in response to second-messenger-linked pathways and direct hormonal regulation. Activators and repressors interact with DNA binding sites to up- and downregulate *aqp* gene transcription. The differential regulation of AQP1 and AQP4 expression levels in patients with temporal lobe epilepsy could offer new insights into treatments for the approximately 30% of patients who are resistant to current drug therapies [522]. The transcription of AQP2 is upregulated by a cAMP-responsive promoter element [523]. Ca^2+^-dependent mechanisms of *aqp* gene regulation include the coordinated regulation of AQP2 expression by Ca^2+^-dependent calcineurin and osmotic stress [524]. Steroid hormonal control of expression is illustrated by the upregulation of AQP3 by estrogen, acting at an estrogen-responsive element in the *aqp3* promoter, relevant to increased motility in estrogen-receptor-positive breast cancers [450]. An estrogen response element in the *aqp5* promoter is linked to the enhanced invasion and proliferation of endometrial cells [525]. Cellular responses to environmental stressors show that the modification of AQP expression is an effective component of physiological adaptations. The regulation of AQP expression by osmotic changes is illustrated by a biphasic response after hyperosmotic challenge seen in mouse collecting duct cells, showing an acute decrease followed by an increase in AQP2 abundance [526]. Conversely, a hypotonicity-associated reduction in AQP2 transcription occurs independently of vasopressin [527]. AQP5 transcription is enhanced in lung alveolar epithelial cells after hypertonic challenge and is dependent on the HIF1-α transcription factor [528]. An osmosensory signaling role for AQPs is a logical proposal [529]. Levels of expression are sensitive to other environmental factors, such as hypoxia, oxidative stress, or pH. AQP1 upregulation by hypoxia in mouse endothelial cells could assist in homeostatic rebalancing, based on observations that AQP1 facilitates gas permeation across plasma membranes in some tissues [322].

### 6.2. Control of AQP Trafficking and Subcellular Localization

Endocytosis and exocytosis between plasma membrane and intracellular vesicles play a key role in modulating AQP function in response to receptor-mediated signaling [1]. AQP2 is shuttled to the apical plasma membrane in the kidney collecting duct after phosphorylation by PKA, in response to cAMP synthesis downstream of vasopressin signaling [530,531,532]. A second mechanism promoting AQP2 trafficking is correlated with reduced activity of protein phosphatase and decreased intracellular Ca^2+^, independent of cAMP [533]. Other examples of cyclic-nucleotide-dependent mechanisms of AQP protein regulation include the cAMP-induced translocation of AQP5 to the basolateral epithelial cell membrane in uterine tissue [534]. The hormone glucagon, acting via cAMP, triggers the redistribution of intracellular AQP8 to the plasma membrane, increasing water permeability [384]. AQP8 in rat hepatocytes is intracellular in the resting state, but insertion into the plasma membrane was shown to be stimulated by cAMP [384].

The control of AQP membrane abundance, as exemplified by the case of AQP2 in response to vasopressin, is often seen as an idiosyncrasy, even though hormones may control the localization of other human AQPs [12]. This assumption of constitutive expression has been challenged by work showing that subcellular relocalization in response to non-hormonal signals such as tonicity, oxygen availability, and temperature is a regulatory mechanism controlling human AQP1 [535,536], AQP4 [31,41,537], and AQP5 [538]. Characterizing the molecular mechanisms of AQP subcellular relocalization in vitro and in vivo is now essential to fully understand the physiological control of water homeostasis.

AQPs associated with intracellular organelles are an area of growing research interest. AQP3 in endosomes is needed for immune response development in mice. Adaptive immunity to pathogens starts with the presentation to CD8^+^ T-cells of a foreign antigen, which after ubiquitination and proteasome degradation must be released into the cytosol; this process requires AQP3 thought to act by enabling H_2_O_2_ entry into the endosome to cause lipid peroxidation and release [539]. AQP8 in the inner mitochondrial membrane was shown to confer Hg^2+^-sensitive water permeability in liver cell mitochondria [540]. In a pancreatic β-cell line, excess H_2_O_2_ was linked to a failure of insulin release, potentially significant for diabetes; increased levels of AQP8 had a protective effect, proposed to result from the AQP8-mediated fluxes of H_2_O_2_ across mitochondrial and plasma membranes [541]. AQP8 in the inner mitochondrial membrane of renal proximal cells increases as an adaptive response to acidosis, allowing the excretion of ammonia [542]. The downregulation of AQP11 in the ER impairs the flux of H_2_O_2_, without altering mitochondrial or plasma membrane H_2_O_2_ transport [543]. AQP11 transcript levels are upregulated in the ER of adipocytes during differentiation and lipolysis, and are decreased by proinflammatory agents that promote ER stress [544]. Kidneys of *aqp11*-null mice show impaired glycosylation and trafficking of polycystin-1, resulting in a polycystic kidney phenotype [545].

As detailed above, AQPs have been shown to cluster in growth cones and process extensions during development, where they are necessary for migration. AQPs incorporated into protein signaling complexes (caveolae; ephrin; and cytoskeleton) show that they can function as components of larger signaling effector complexes [546], not only as solitary channels working independently.

### 6.3. Properties of Mammalian AQP Ion Channels

Reconstituted in lipid bilayers, AQP0 ion channels carry both cations and anions, but show a higher permeability for anions [7,20,547,548]. AQP0 water permeability is modulated by pH and Ca^2+^ through distinct amino acids. In loop A, His40 is responsible for pH sensitivity, and the phosphorylation of the C-terminus Ser235 regulates Ca^2+^ sensitivity by modulating the CaM affinity of a nearby CaM-binding domain [189,549].

Although the activation of AQP1 ion conductance and the ion functionality were initially controversial, AQP1 is now accepted as a cyclic-nucleotide-gated ion channel [21,22,23]. The AQP1 ion channel function has provided new perspectives on its physiological and pathophysiological roles [45,550]; however, the presence of cGMP alone does not ensure ionic conductance through AQP1 and depends on additional regulatory mechanisms, such as the phosphorylation of a tyrosine residue (Tyr253) in the C-terminus of human AQP1 [24]. cAMP activates AQP1 indirectly via a kinase-dependent mechanism [21,23]. Human AQP1 cation currents are elevated by the phosphorylation of two threonine residues (Thr157, Thr239) by protein kinase C [551], which also was shown to induce the relocalization of AQP1 to the plasma membrane in HEK293 cells [535], suggesting another potential mechanism for increased current amplitudes.

The third mammalian AQP currently known to facilitate ion conductance when expressed in oocytes is AQP6 [26]. AQP6 is active under acidic conditions (pH < 5), consistent with its proposed role in intracellular vesicles in epithelial cells of the kidney [26,47,75,552,553]. AQP6 exhibits an N-terminal CaM-binding site [554]. The effects of CaM binding on channel function or subcellular localization are unknown, though there is a characterized intracellular retention signal in the N-terminus [552]. The cGMP binding feature of AQP1, an arginine-rich loop D, is not present in AQP6. AQP6 water permeability is low [48], but AQP6 ion and water permeabilities have been shown atypically to be activated by mercuric chloride, HgCl_2_ [26,47]; in contrast, other AQPs are inhibited by the binding of mercury to a cysteine residue (Cys189 in human AQP1) near the intrasubunit pore mouth [555,556].

AQP6 is mainly selective for anions, including nitrate and chloride (NO_3_^−^, Cl^−^), and is pH-sensitive, with elevated water and ion permeabilities detected at an acidic pH < 5.5 [26]. Loop B in AQPs connects the second and third transmembrane domains and contributes to the intrasubunit pore, a domain associated with AQP water channel function [557]. Both water and ions in AQP6 are thought to permeate through the intrasubunit pores, not the central pore, since sites in loop B have been observed to be crucial for ion conductance properties [26,75], and studies on the Hg^2+^ binding kinetics support a model consistent with four ion pathways per AQP6 tetramer [47]. Two amino acid residues, tyrosine (Tyr37) and threonine (Thr63), located opposite the NPA motif are thought to be important for the AQP6 anion conductance; a reduced nitrate permeability has been observed in AQP6 channels lacking these residues [75]. Moreover, the substitution of an asparagine residue (Asn60) in the second transmembrane domain, unique to AQP6, with glycine (characteristic of other mammalian AQP channels) restricted the anion conductance but enhanced water permeability [48], suggesting that the rigidity of the channel influences the passage of larger substrates, such as nitrate, versus smaller water molecules [48].

### 6.4. Overview of Pharmacological Tools

The targeted modulation of AQPs offers an opportunity for developing treatments with potential value in diverse clinical conditions [558]. Pharmacological blockers for members of the orthodox class of AQP channels (Table 2) are thought to work primarily by occluding the permeation pathway at either the extracellular or the intracellular side of a subunit pore. The classic extracellular inhibitors are mercuric compounds that block a subset of AQPs by the covalent modification of a cysteine residue in the loop E domain. The first non-mercurial blocker discovered for AQPs was the quarternary ammonium ion tetraethylammonium (TEA), which when applied extracellularly caused a reversible partial inhibition of osmotic water permeability in AQP1-expressing *Xenopus laevis* oocytes, later extended to include some other AQPs [559,560,561]. However, no therapeutic value of TEA was evident based on several considerations: the dose-dependent inhibitory effect saturates at levels that still allow substantial residual water flux; TEA is not selective for AQPs but also blocks other channels; and water channel inhibition by TEA has not been replicated in all the systems tested [562], possibly because the residual water flux levels are not rate-limiting, the blocking effect is directional, or differences in glycosylation or other factors affect channel susceptibility to TEA. Other potentially drug-like extracellular AQP1 inhibitors characterized by Seeliger and colleagues were confirmed in oocyte expression assays and red blood cells to show IC_50_ values ranging 8 to 18 µM [563].

Intracellular inhibitors of AQP channels in the orthodox class are based on analogs of the arylsulfonamide loop diuretic drugs, furosemide, and bumetanide that have been used clinically for decades to lower fluid volume and reduce blood pressure. The bumetanide derivative AqB013 was characterized as a dose-dependent blocker of water flux in AQP1 and AQP4 channels expressed in oocytes; a second bumetanide derivative, AqB011, was found to selectively block the ion channel activity of AQP1 mediated by the gated central pore; and a furosemide derivative, AqF026, was discovered to be an agonist of AQP1 and AQP4 water channel activity [430,566,574]. A reported lack of success in replicating the drug formulation methods and activities of arylsulfonamide compounds [575] suggests that further work with chemically verified structures and in a variety of AQP1 expression systems will be needed to clarify the basis for differences in outcomes.

The sulfonamide compound acetazolamide has been characterized as an inhibitor of orthodox AQPs, found to block AQP1 (IC_50_ 5.5 µM) [563] and AQP4 (IC_50_ 0.9 µM) [568], though multiple reports of its effective use as a blocker must be balanced against findings challenging the inhibitory effect [576]. N-(5-sulfamoyl-1,3,4-thiadiazol-2-yl) acetamide is a derivative of acetazolamide, proposed as a candidate AQP4 antagonist [577]. Ethoxyzolamide, another carbonic anhydrase inhibitor, was reported to inhibit water transport of the M23 isoform of human AQP4 at 20 μM in oocytes [568]. TGN-020 has been characterized as a selective AQP4 blocker (IC_50_ 3.1 µM). Delivered intraperitoneally at 100 mg/kg, TGN-020 improved locomotor recovery and reduced edema (though only by about 3%, as compared with untreated injured animals) after spinal cord compression injury in rats [578]. Patil and colleagues identified six new AQP1-blocking compounds, classified as aromatic sulfonamides and dihydrobenzofurans with IC_50_ values ranging from 3 to 30 µM, using high-throughput screening in calcein-loaded AQP1-expressing CHO cells, confirmed in proteoliposomes and erythrocyte ghosts [579]. A variety of other agents have also been proposed as AQP modulators, though published evidence with independent confirmations remains limited thus far. Topiramate, suggested to inhibit AQP4 [571], is a sulfamate-substituted monosaccharide. IMD-0354 (N-(3,5-bis(trifluoromethyl)phenyl)-5-chloro-2-hydroxybenzamide) (also called AER-270), known as a cardioprotective agent, is a proposed AQP4 antagonist [580]. Antiepileptic agents such as phenytoin, lamotrigine, and topiramate, investigated with in silico and in vitro screening, are suggested to inhibit AQP4 [571].

Agents isolated from traditional medicinal herbs are a promising source of new agents [68]. Identified compounds from *Bacopa monnieri* were characterized as selective blockers of AQP1 but not AQP4 channels expressed in oocytes [222]. Bacopaside II demonstrated impressive therapeutic benefits in a mouse model, reducing stress-induced hypertrophic remodeling of the heart by blocking the AQP1-mediated transport of H_2_O_2_ [49]. An agent found in various foods such as Manuka honey, 5-hydroxymethyl furfural, blocks the AQP1 ion channel without altering water flux [429].

The eventual identification of clinically useful ‘aquaretics’, as selective blockers of AQP water pores, is highly anticipated as an exciting advance for the therapeutic management of edema and fluid volume disorders. Looking beyond the role of AQPs as water channels, blockers of AQP dual water and ion channels have promise as tools for reducing metastasis in diseases, such as cancer, that depend on AQP-enhanced cell motility [581]. Blockers of AQP-mediated H_2_O_2_ signaling pathways in particular offer intriguing opportunities as candidates for interventions in diabetes, kidney disease, cardiac disease, and immune system dysfunction.

## 7. Future Directions for AQP Research

AQP-mediated water and solute transport are crucial for the correct functioning of a variety of organ systems and cell types. The structural biology of the transmembrane domains of the AQP family is well-understood, with over 40 structures currently deposited in the Protein Data Bank. However, the structures of the intracellular AQP termini and the quaternary structures of AQP-containing complexes are poorly understood. Recent developments in the solubilization of membrane proteins [582] and cryo-electron microscopy for structural biology [583] will likely support progress in this area.

AQPs support whole-body water homeostasis via the kidneys, and it is in this context that the physiological role of AQPs is best understood. In particular, the reabsorption of water through AQP1 in the proximal tubule and the vasopressin-induced reabsorption of water through AQP2 in the collecting duct are considered prototypes of constitutive and regulated AQP function, respectively. The regulatory sub-cellular relocalization triggered by environmental changes offers intriguing new possibilities to develop AQP-targeted therapeutics, especially in edema treatment. AQPs also support a variety of secretory systems, including the production of CSF, saliva, tears, and bile, and are attractive drug targets for conditions involving under- or overproduction of these fluids.

AQPs are expressed throughout the ocular and auditory systems; aside from AQP0 in the lens, their exact roles are yet to be full elucidated. Similarly, the role of AQPs in the peripheral nervous system is controversial, but if validated could provide a route to the development of novel analgesics.

AQPs have key functions in the maintenance of several physiological barriers, including the skin and BBB, as well as supporting endothelial health. AQPs are expressed in a variety of immune cells, but exactly whether or how they support immune function is an open question.

AQPs have a key role in supporting cell migration, and this is best understood in the contexts of angiogenesis and cancer cell migration. The aberrant expression of AQPs in many cancer types presents new opportunities for the development of new anti-metastasis and possibly anti-proliferative therapies, as well as diagnostic biomarkers.

New pharmacological tools will be crucial to support all this ongoing research [1]. A deeper understanding of the signaling pathways that control AQP activation, gating, and trafficking will be vital for the development of AQP-targeted therapies and research tools, and it is crucial for the field to begin to understand how and why the effects of many current putative AQP pore-blocking drugs are so difficult to reproduce between different laboratories and expression systems. Combinations of therapeutic agents that target multiple components of an interacting set of signaling proteins could be used to focus treatment effectiveness and tailor therapies by cell type, thus reducing side effects [68]. The field of AQP pharmacology is still young but has the potential for major breakthroughs, with much remaining to be discovered.

## Figures and Tables

**Figure 1 ijms-23-01388-f001:**
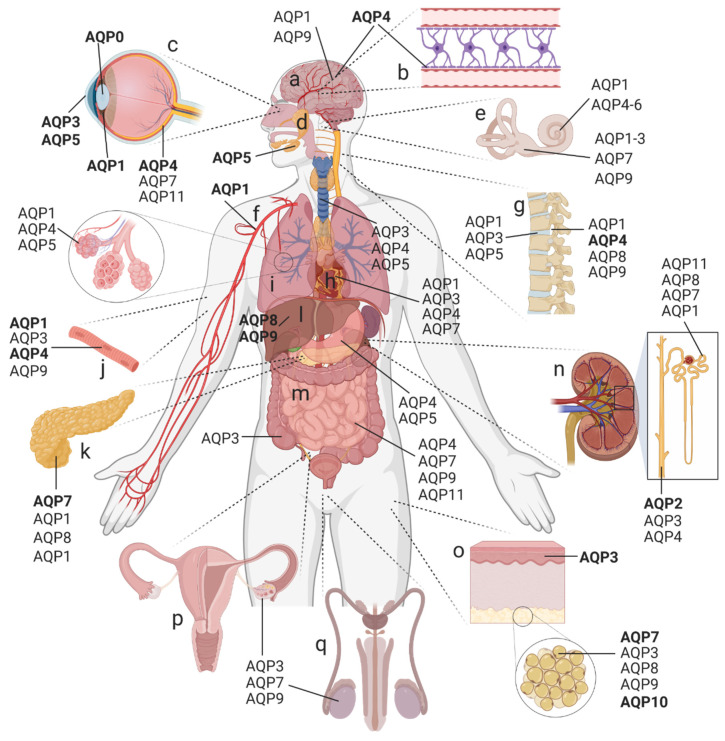
AQP distribution in the human body. Expression of AQP paralogs in the (**a**) brain, (**b**) blood–brain barrier, (**c**) eye, (**d**) exocrine glands, (**e**) inner ear, (**f**) cardiovascular system, (**g**) spine, (**h**) heart, (**i**) respiratory tract (trachea and lung; inset showing alveoli), (**j**) skeletal muscle, (**k**) pancreas, (**l**) liver, (**m**) gastrointestinal tract, (**n**) kidney, (**o**) skin (inset showing adipose tissue), and (**p**) female as well as (**q**) male reproductive tracts. This summary is not comprehensive; minor AQP subtypes are omitted for clarity. **Bold text** is used to highlight the major AQPs studied in the selected tissues. Created with BioRender.com; adapted from Day et al., 2014 [12].

**Figure 2 ijms-23-01388-f002:**
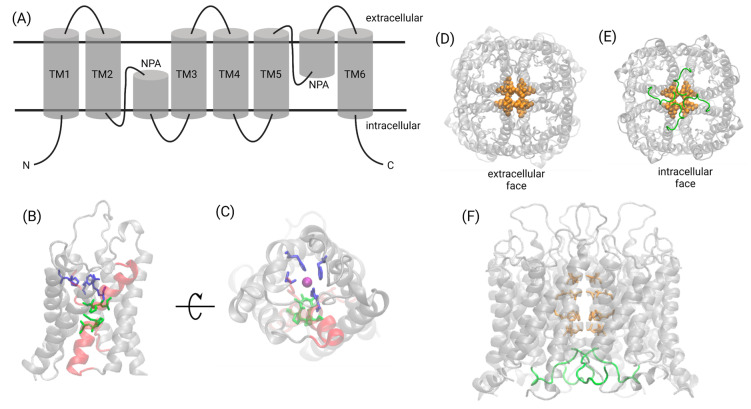
Structural biology of the AQP family. (**A**) The signature fold of the AQP family consists of six transmembrane helices and two helix-forming re-entrant loops containing the signature NPA motif. (**B**,**C**) Water transport and selectivity is facilitated by the NPA motifs (green) found at the interface of the two helical re-entrant loops (red) and the aromatic/arginine selectivity filter (blue). Water molecules (a single water oxygen at the selectivity filter is indicated by a purple sphere) traverse the pore in single-file. (**D**–**F**) The central pore formed at the fourfold axis of AQP1 contains two rings of bulky hydrophobic residues (orange) that prevent pore hydration in the absence of a cGMP signal. cGMP binding at loop D (green) activates the ion channel. Created with Biorender.com.

**Figure 3 ijms-23-01388-f003:**
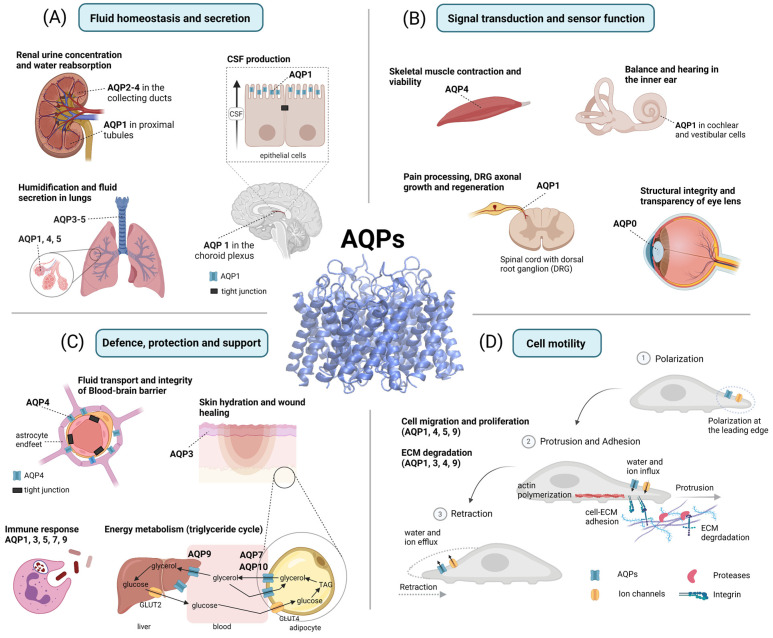
Functional roles of AQPs. (**A**) *Fluid homeostasis and secretion*: In the kidney, AQP1 regulates water reabsorption in the proximal tubules, while AQP2–4 are involved in urine concentration. In the central nervous system (CNS), AQP1 is involved in cerebrospinal fluid (CSF) production in the choroid plexus. In the lungs, AQPs facilitate transendothelial and transepithelial water flow. (**B**) *Signal transduction and sensor function*: AQP4 is involved in skeletal muscle contraction and viability. In the spinal cord, AQP1 is thought to contribute to pain processing and promote axonal growth as well as the regeneration of dorsal root ganglia (DRG). In the inner ear, AQPs are involved in balance and hearing. In the eye, AQP0 facilitates the structural integrity and transparency of the lens. (**C**) *Defense, protection, and support*: AQP4 is involved in blood–brain barrier (BBB) integrity, astrocyte plasticity, glial scar formation, and cerebral waste clearance. AQP3 supports skin hydration and wound healing. AQP1, 3, 5, 7, and 9 are involved in immune cell activation and pathogen elimination (phagocytosis). AQP7, 9, and 10 are involved in the glycerol transport that supports energy metabolism. (**D**) *Cell motility*: AQP1, 4, 5, and 9 are polarized at the leading edge of migrating cells and are thought to promote the cellular migration stages of polarization, protrusion, adhesion, and retraction. Additionally, AQP1, 3, 4, and 9 are assumed to enhance the degradation of the extracellular matrix (ECM). Created with BioRender.com.

**Table 1 ijms-23-01388-t001:** AQP classification with permeant substrates and main sites of expression. Chromosome location and water permeability data adapted from [69]. Asterisks (*) highlight permeability results that are controversial or not yet accepted as readily reproducible.

Aquaporin	Chromosome	Water Permeability (P_f_) [×10^−14^ cm^3^ s^−1^]	Permeability to Molecules Other Than Water	Main Expression Sites
Orthodox (classical) AQPs				
AQP0	12q13	0.25	Ions [19,20]	Eye lens
AQP1	7p14	6.0	Monovalent cations [24,36,42], nitric oxide [70], H_2_O_2_ [49,55], and glycerol * [71]	Central nervous system (CNS), inner ear, eye, kidney, endothelium, lung, skeletal muscle, cartilage, and erythrocytes
AQP2	12q13	3.3	None known	Kidney, inner ear, and reproductive tract
AQP4	18q22	24	Nitric oxide [72]	CNS, inner ear, retina, kidney, gastrointestinal tract (GIT), lung, and skeletal muscle
AQP5	12q13	5.0	H_2_O_2_ [51]	Secretory glands, inner ear, eye, kidney, GIT, and lung
AQP6	12q13	Low; no quantitative data	Ammonia [73], glycerol, urea [74], nitrate [75], and anions (NO_3_^−^, Cl^−^) [76]	Inner ear, kidney
AQP8	16p12	No quantitative data	Urea, ammonia, and H_2_O_2_ [77]	Liver, kidney, adipose tissue, pancreas, GIT, and reproductive tract
Aquaglyceroporins				
AQP3	9p13	2.1	Glycerol [78], H_2_O_2_ [9], urea * [78], and ammonia [79]	Skin, inner ear, eye, adipose tissue, kidney, GIT, heart, lung, reproductive tract, and cartilage
AQP7	9p13	No quantitative data	Arsenite [80], glyerol and urea [81], and ammonia [82]	Adipose tissue, pancreas, liver, kidney, inner ear, GIT, heart, reproductive tract
AQP9	15q22	No quantitative data	Arsenite [80], carbamides, polyols, purines, pyrimidines [83], ketone bodies [84], lactate [85], ammonia [86], glycerol, urea [83,87,88], and H_2_O_2_ [54]	Liver, adipose tissue, CNS (unclear for humans), inner ear, and reproductive tract
AQP10	1q21	No quantitative data	Glycerol [89]	Adipose tissue and reproductive tract
Unorthodox AQPs/S-aquaporins				
AQP11	11q13	~2	Glycerol [29,90,91]	Retina, kidney, GIT, and reproductive tract
AQP12	2q37	No quantitative data	Unknown	Pancreas

**Table 2 ijms-23-01388-t002:** Overview of pharmacological agents for aquaporin inhibition. Proposed AQP inhibitors are listed with chemical structures (created with ChemSketch), evidence for inhibitory effects on AQP paralogs (h, human; m, mouse; and r, rat), and evaluation of their current pharmacological value. The experimental expression system that was used to demonstrate inhibition is indicated as (S), in silico; (P), proteoliposome; (O), Xenopus laevis oocyte; (M), mammalian cell line; or (I), in vivo.

Agent	Structure	Evidence for AQP Inhibition	Pharmacological Value
Tetraethyl-ammonium (TEA)	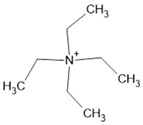	hAQP1/2/4 (O, M) [559,560,561]	Low potency; no AQP selectivity; and TEA inhibition is not reproducible in all assay systems
Bumetanide(anti-diuretic) AqB013	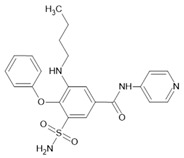	h/rAQP1/4 (O) r/mAQP4 (I) [430,564,565]	Reproducible effects in other systems remain to be verified
Furosemide(anti-diuretic) AqF026	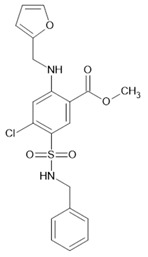	Increased hAQP1 activity [O, I] [566]	AQP1 specificity; reproducible effects in other systems remain to be verified
Sulfonamideacetazolamide	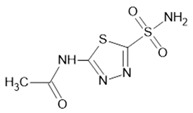	rAQP1/4 (O, M, and P) [563,567,568,569]	Controversial
TGN-020	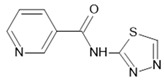	hAQP4/1 (S, O) [570,571] rAQP4 (I) [572,573]	Controversial
Anti-epileptic topiramate	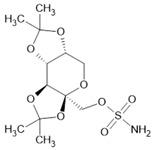	hAQP4 (O, S) [571]	Reproducible effects in other systems remain to be verified
Medical herb compound bacopaside II	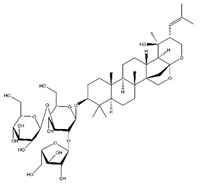	hAQP1 (O, M) [222] mAQP1 (I) [49]	AQP1 selectivity, possible application in H_2_O_2_ flux blockage in treatment of cardiac diseases

## Data Availability

Not applicable.

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
