# Peer review of "Signaling Mechanisms and Pharmacological Modulators Governing Diverse Aquaporin Functions in Human Health and Disease"

_ijms, 2022, doi:10.3390/ijms23031388_

Round 1
Reviewer 1 Report
The current Manuscript extensively discussed about the Aquaporin biology (structure, function and localization in several organs throughout the human body) and small molecules targeting the inhibition of AQPs and their effects. Authors have extensive background in AQP biology and reveled the key roles of all AQPs in water balance, glandular secretions, signal transduction and sensation, barrier function, immunity and inflammation, cell migration, and angiogenesis. However, authors need to address fallowing minor comments, which needs to be addresses before the manuscript accepted in IJMS.
Comments:
1. Can you also include the paragraph highlighting or discuss regarding the diseases associated with mutations and organ dysfunction of AQPs in humans? In addition is there any animal models that indicate the over expression or under expression of (excluding the kidney) AQPs in peripheral organs lead to disease severity and also in cancer progression?
2. Is there any role of AQPs across the intestine (duodenum, Jejunum and ileum, Colon) ?. Also, any Disease associated with dysfunction of AQPs in the gut? Needs to include.
3. In section 6 (Modulation of AQP channel activities), sub-section 6.1 and 6.2 needs be deleted, as both parts are out of scope from current subject. Also Signals or secondary messengers cAMP, cGMP and Ca2+ more relevant to physiological property of AQPs, may not corresponding to direct pharmacological inhibition.
Author Response
We thank the reviewers for careful analyses and valuable comments on our MS. The MS has been revised to address all the concerns raised, as summarized below and shown in the pdf version of the revised MS with track-changes.
Reviewer 1 had minor comments.
- Include a paragraph highlighting or discuss regarding the diseases associated with mutations and organ dysfunction of AQPs in humans. Are there any animal models that indicate the over expression or under expression of (excluding the kidney) AQPs in peripheral organs lead to disease severity and also in cancer progression?
These topics were embedded throughout the different sections of the MS, in which we detailed both the physiological roles and pathological consequences of AQPs in organ systems. To provide a broader perspective as requested by Reviewer 1, we selected a set of recent review articles, now listed in the final sentence of section 1.3 (pg 8).
- Is there any role of AQPs across the intestine (duodenum, jejunum and ileum, colon); any disease associated with dysfunction of AQPs in the gut?
We thank the reviewer for this suggestion. We have added a new section 2.4 titled "Secretion of gastrointestinal fluids in the digestive system" covering these topics (pg 13).
- In section 6 (Modulation of AQP channel activities), sub-section 6.1 and 6.2 needs be deleted, as both parts are out of scope from current subject. Signals or secondary messengers cAMP, cGMP and Ca2+ more relevant to physiological property of AQPs, may not corresponding to direct pharmacological inhibition.
We apologize for the confusion. In the revised version, we re-titled section 6 as "Physiological and pharmacological modulation of AQP channel activity" in order to emphasize that this section intentionally includes both endogenous regulatory pathways and exogenous pharmacological agents (pg 27).
Reviewer 2 Report
Dear Authors,
It was for me great pleasure to review your manuscript entitled “Signaling mechanisms and pharmacological modulators governing diverse aquaporin functions in human and diseases” (Manuscript ID: ijms1531001). These comprehensive and very interesting review paper summarize the present knowledge about the role of AQPs in mammalian organism and shed a new light on signaling mechanisms, pharmacological modulators and novel, potential treatments for the AQP-associated disorders. In my opinion this manuscript fulfills the requirements set for publication in International Journal of Molecular Sciences and I recommend publishing it after minor revision. Before the final decision of the publication, please consider the following remarks:
- In my opinion according to new nomenclature it should be avoided to use the term of “isoform”. AQPs are encoded by different genes and they are therefore paralogs belonging to the aquaporin superfamily. For example, description of the Figure 1.
- In the Box 1. Abbreviations used in this review. There is no explanation of DRG.
- Figure 1 resembles to a high extend Figure 1 from PMID: 24090884. Maybe it will be properly to state in the figure legend that drawing is “Modified from…” Otherwise, this can be considered a form of autoplagiarism.
- Figure 1 summarize distribution of AQPs in the human body. On the page 11 you indicate that in the human kidneys are located nine AQPs. There is lack of references for this fact in the text. It is also contradictory with the figure 1, where are presented only 7 AQPs. There is lack of AQP5 and AQP6. It is generally known that in mammalian kidneys are located nine AQPs (PMID: 31682170). But for all I know in available literature still there is no direct evidences for AQP6 in human kidney.
- Figure 1 present only 3 aquaporins in male reproductive system. According to present literature and studies conducted in animals and humans as many as 11 out of 13 mammalian aquaporins have been identified (PMID: 34810284; PMID: 33338597). In human were identified also AQP1, AQP4, AQP8 and AQP11. I know that it is difficult to present all AQPs located in particular organs of human body, but in this form the figure may mislead the reader. Maybe it will be much better to remove bold text and just leave in description that figure presents the most studied AQPs in human.
- It is known that AQP2 is selectivity permeable only for water. But empty place in table 1 may confuse the inexperienced reader. In my opinion it will be better to add some information about it.
- It is known that AQP2 is located in mammalian vas deferens and vagina (PMID: 33338597). In my opinion it should be add in the table 1 information that this protein is also present in reproductive tract.
- Table 1. You missed that AQP3 is expressed in kidney.
Best regards,
Author Response
Reviewer 2 provided excellent suggestions and corrections that were greatly appreciated. The minor concerns raised and our responses are summarized below.
(1) The first point focused on nomenclature and correct use of the terms "paralog" and “isoform”.
We are grateful for the advice. The terminology corrections have been made throughout the text, and in the legend to Figure 1.
(2) The reviewer correctly noted that in Box 1, there was no explanation of DRG.
The DRG acronym and definition have been added to Box 1.
(3) The Figure 1 legend should acknowledge that the illustration was modified from PMID: 24090884.
This detail has been added to the revised Fig 1 legend.
(4) The statement that human kidneys have 9 AQPs was not referenced, and was contradicted by Fig 1 showing only 7 kidney AQPs.
A reference for the kidney expression has been added in the text, and the Figure legend has been edited to explain that minor classes of AQPs have been omitted from the drawing for clarity.
(5) The reviewer observed that Figure 1 presented only 3 aquaporins in male reproductive system though more have been identified, and suggested that the figure might be more clearly described as covering 'only the most studied AQPs'.
This good suggestion was implemented. The Figure 1 legend now states that not all AQPs are illustrated in the diagram; minor classes have been omitted for clarity.
(6) In Table 1, empty space for AQP2 showing permeability only for water might confuse the inexperienced reader. AQP2 should be listed for mammalian vas deferens and vagina, and AQP3 in kidney.
All these details have been added to Table 1 as suggested.